# Rapid Tartrazine Determination in Large Yellow Croaker with Ag Nanowires Using Surface-Enhanced Raman Spectroscopy

**DOI:** 10.3390/nano8120967

**Published:** 2018-11-23

**Authors:** Jia Song, Yuanyi Zhang, Yiqun Huang, Yuxia Fan, Keqiang Lai

**Affiliations:** 1College of Food Science and Technology, Shanghai Ocean University, Shanghai 201306, China; susanjia123@163.com (J.S.); jennifer.yuanyizhang@foxmail.com (Y.Z.); yxfan@shou.edu.cn (Y.F.); 2School of Chemistry & Biological Engineering, Changsha University of Science & Technology, Changsha 410076, Hunan, China; yiqunh@csust.edu.cn; 3Engineering Research Center of Food Thermal Processing Technology, Shanghai Ocean University, Shanghai 201306, China

**Keywords:** SERS, Ag nanowires, tartrazine, large yellow croaker

## Abstract

In this work, surface-enhanced Raman spectroscopy (SERS) technology coupled with Ag nanowires was shown to be a promising tool in the detection of tartrazine in large yellow croaker for the first time. Ag nanowires with a uniform diameter were fabricated by an efficient and manageable polyol method. The partial least square model was established for the quantitative analysis of tartrazine, which showed a relatively high linear correlation between actual and predicted concentrations of standard tartrazine solutions. An optimal sample preparation method was also selected and used to extract tartrazine from large yellow croaker within 20 min. The lowest concentration detected was 20.38 ng/cm^2^, which fully meets the requirements of tartrazine testing in aquatic products. This study indicated that SERS technology combined with the as-prepared Ag nanowires could detect tartrazine sensitively and provide an easily operable and time-saving way to monitor tartrazine in large yellow croaker.

## 1. Introduction

Tartrazine is a well-known synthetic colorant that is frequently and widely used in foods (such as soft drinks and candies), pharmaceuticals and cosmetics to make them appealing to customers [1]. The acceptable daily intake of tartrazine in food, proposed by the FAO and WHO, is 7.5 mg/kg. Tartrazine has low toxicity, but its excessive ingestion can result in allergy, asthma, migraine and hyperactivity [2]. It may also trigger sleep disorders in children [3]. Given the potential risks to humans, tartrazine has been banned in countries including Norway and Austria [4]. Increasing concern has been paid to health issues provoked by food additives used in China; the types of foodstuff and the maximum amount of additives allowed have been strictly set by Chinese regulation (GB2760-2014) [5]. Tartrazine, a coloring agent used on yellow croaker (*Pseudosciaena crocea*), in tsukemono and in wine, has been banned by the Ministry of Health of the Republic of China [6]. However, it is still used illegally and in excess in various food products due to its high economic benefits. Hence, developing a time-efficient, cost-effective and sensitive method to detect tartrazine in foodstuff is necessary.

Currently, numerous methods have been developed to analyze tartrazine in different foods. These methods include high-performance liquid chromatography (HPLC) [7], liquid chromatography-tandem quadrupole mass spectrometry [8], absorption spectrometry [9], voltammetry [10] and thin-layer chromatography [11]. Although some of these analytical methods have high accuracy, good reproducibility and reach the necessary level of quantification according to regulations, they also have disadvantages, such as high cost, inconvenient operation, and complicated sample preparation. Surface-enhanced Raman scattering or surface-enhanced Raman spectroscopy (SERS) is one of the most effective analytical tools for detecting the trace components of food owing to its high sensitivity, quick determination and convenient operation. A growing number of research publications on the detection of illegal additives [12], pesticide residues [13,14,15,16] and veterinary drugs in foodstuff are available [17,18]. There are some reports on the use of SERS for the detection of tartrazine [19,20,21] but the number of publications on the determination of tartrazine in foodstuff on the basis of the SERS method is relatively low. In 2016, SERS coupled with Au nanodumbbells was used to analyze tartrazine in drinks [22].

To the best of our knowledge, no report using SERS to detect tartrazine in large yellow croaker has been published. The main objective of this study is to investigate and develop a time-efficient and cost-effective method of detecting trace amounts of tartrazine in large yellow croaker using Ag nanowires.

## 2. Materials and Methods 

### 2.1. Materials and Reagents

AgNO_3_ (>99%), polyvinyl pyrrolidone (PVP) (Mw = 55,000 g/mol) and NaCl (≥99.999%) were bought from Sigma-Aldrich (St. Louis, MO, USA). Glycerol (American Chemical Society grade) was obtained from Aladdin Industrial Corporation (Shanghai, China). Ethanol (analytical grade) was received from Sinopharm Chemical Reagent Co., Ltd. (Shanghai, China). Tartrazine (≥99.0%) and methanol (HPLC grade) were purchased from Sigma-Aldrich (St. Louis, MO, USA). MgSO_4_ was purchased from Aladdin Industrial Corporation (Shanghai, China). Large yellow croaker were bought from a supermarket in Shanghai and we confirmed an absence of tartrazine by the HPLC method, which was performed in the China General Chamber of Commerce Supervision & Testing Center for Food (Shanghai, China). The 0.45 μm polytetrafluoroethene (PTFE) microporous films were obtained from Anpel Corporation (Shanghai, China).

### 2.2. Preparation and Characterization of Ag Nanowires

Ag nanowires were synthesized using the polyol method referred to in our previously published article [23]. Ag nanowires were fabricated by reducing AgNO_3_ in glycerol at a rising temperature. PVP, along with the mixture of NaCl, ultrapure water and glycerol, was also added to the reaction liquid; PVP was the stabilizing agent or surfactant. Then, the cooled products were washed by ultrapure water and ethanol aqueous solution. The UV-Vis absorbance spectrum of the as-fabricated Ag nanowires was recorded by a UV-Vis spectrometer (UV3000PC, MAPADA Instruments Ltd., Shanghai, China). Transmission electron microscopy (TEM; JEM-2100F, JEOL Ltd., Tokyo, Japan) was used to analyze the morphology and sizes of the synthesized samples.

### 2.3. Preparation of Standard Solution

Tartrazine powder was dissolved in ultrapure water to prepare the standard stock solution of 100 μg/mL and this was stored at 4 °C. Then, a series of working standard solutions was obtained by a consecutive dilution method with the concentrations ranging from 10 ng/mL to 10 μg/mL.

### 2.4. Pretreatment of Large Yellow Croaker Sample

The large yellow croaker skin was cut into small circular pieces of 0.5 cm in diameter; we then measured the weight of each piece. A total of 200 μL of tartrazine solution with different concentrations were dropped onto the small pieces of skin to cover the surface. Next, stewing was performed. Subsequently, the large yellow croaker skin was transferred to the centrifuge tubes and 4 mL of ultrapure water was added. Then, the mixture was sonicated for 5 min.

To optimize sample preparation and reduce time and cost, we compared three methods in further experiments. Extractions were filtered through a 0.45 μm PTFE microporous film and then the filtrates underwent SERS analysis. This method was performed only with water (W) extraction and the samples directly underwent a type of filtration purification called the W method. For the second method, 1 mL of methanol (M) was added to the deposit protein. After the ultrasound step, the vortexed solution was filtered with a PTFE microfilter; this was called the W+M method. The third method, called the W+M+C method, had the same steps as that of W+M method, except that a further purification of centrifugation (C) at 5000 rpm for 6 min was performed before filtration.

### 2.5. Raman and SERS Measurements

The normal Raman spectrum of tartrazine and the SERS spectra of standard tartrazine solution or extracts were collected using a Nicolet DXR microscopy Raman spectrometer (Thermo Fisher Scientific Inc., Waltham, MA, USA) equipped with a 633 nm He–Ne laser source. A 20× objective at 5 mW laser power was used throughout the experiments.

For the normal Raman spectrum of tartrazine, a small amount of tartrazine was deposited onto a glass slide and squeezed to a thin film before its normal Raman spectrum was recorded.

For SERS spectra collection, samples were prepared by mixing 20 μL of Ag nanowires colloidal solution, 80 μL of standard tartrazine solution or large yellow croaker skin extract and 10 μL of MgSO_4_ solution (150 mM). The selection of the volume ratio between the Ag nanowires substrate and tartrazine solution, as well as the types (sulfate, chloride, perchlorate) and amount of aggregation agent, were based on our preliminary experiments. Then, 5 μL of aliquot was allowed to evaporate under ambient conditions for SERS detection. The final spectrum was the average of 10 spectra collected from 10 different random positions. This process was repeated in quadruplicate for reproducibility. 

### 2.6. Data Analysis

Delight, version 3.2.1, (DSquared Development Inc., LaGrande, OR, USA) was used for the analysis of spectral data. For quantitative analysis, the partial least square (PLS) model was established to predict the analyte concentrations.

The coefficient of determination (R^2^), the root-mean-square error (RMSE) and the ratio of performance to deviation (RPD) were used to evaluate the model. The higher the R^2^ value (the closer to 1) or the lower the RMSE value, the better predictability the model has [24].

## 3. Results

### 3.1. Spectral Features of Tartrazine Dye

Figure 1 shows the Raman spectrum of tartrazine solid. The peaks at 485, 618, 638, 1129, 1358, 1477, 1503 and 1599 cm^−1^ were easily recognizable. Among these peaks, the most prominent tartrazine peak at 1599 cm^−1^ was assigned to the in-plane bending of OH, the asymmetric stretching of COO^−^, the overlapping effect of symmetric stretching and the out-of-plane C–H deformation of two phenyl rings. The other major peak at 1358 cm^−1^ was attributed to the rocking of the phenyl ring, the symmetric stretching of COO^−^ and the stretching vibrations of –C–N=N–C–. Table 1 shows the assignments of their vibrational bands [25].

### 3.2. Ag Nanowire Characterization 

The UV-Vis absorbance spectrum of the as-prepared Ag nanowires exhibits two characteristic absorption peaks at 349 nm and 380 nm, as shown in Figure 2a. The peak located at 349 nm is ascribed to the out-of-plane quadrupole resonance of Ag nanowires while another peak with high intensity at 380 nm is attributed to the out-of-plane dipole resonance of Ag nanowires [26]. 

To determine the normal distribution and a certain amount of homogeneity, we carried out TEM tests. The TEM images of Ag nanowires are shown in Figure 2b. The figure shows that the diameter of the as-prepared Ag nanowires was relatively uniform. An average diameter of 49.4 ± 3.9 nm and length of 7–10 μm were observed by calculating 100 nanowires. 

### 3.3. SERS Analysis of Tartrazine 

Figure 3 shows the SERS spectral features of the standard tartrazine solutions. The characteristic peaks of standard tartrazine solution can be clearly distinguished at a concentration as low as 0.01 μg/mL. The limit of detection, calculated based on HPLC, was 5.2 ng/mL [27], which was close to the minimum concentration visually observed by our SERS method.

The strongest peak of tartrazine was still visible at 1599 cm^−1^. However, some peaks may red shift or blue shift compared with the normal Raman spectrum shown in Figure 1. For instance, the characteristic peak at 1503 cm^−1^ in the Raman spectrum of tartrazine red shifted to 1504 cm^−1^ in the SERS spectra. The characteristic peak at 1477 cm^−1^ in the Raman spectrum of tartrazine red shifted to 1474 cm^−1^ in the SERS spectra. The shifts in the characteristic peaks were due to changes in site and adsorption orientations or to changes in polarizability after interactions between the analyte and substrate [28,29]. The as-prepared Ag nanowires as SERS substrate could greatly enhance Raman scattering signals of tartrazine. Based on the method described by Ru et al. [30], the enhancement factors for tartrazine were calculated as 3.9 × 10^6^ (based upon the strongest peak at 1599 cm^−1^), the reproducibility of the as-prepared Ag nanowires was reported in our previous study [23].

As shown in Figure 3, the intensity of the characteristic peaks increased remarkably with increasing concentration. Therefore, the quantitative analysis model of the standard tartrazine solutions may be established based on the SERS spectra. The SERS spectra of tartrazine solutions at eight different concentration levels (10 ng/mL–10 μg/mL) were collected to establish the PLS model (*n* = 32, four spectra for each concentration). As shown in Figure 4, a high linear correlation (R^2^ = 0.970) was observed between the actual and the predicted concentrations. High RPD (6.10) and low RMSE (0.26) values also demonstrate that this model has relatively good predictability [16].

### 3.4. SERS Analysis of Tartrazine in Large Yellow Croaker

To simulate the process of the illegal addition of tartrazine to the large yellow croaker, we experimentally spiked large yellow croaker peel extracts with tartrazine (Figure 5a). We aimed to analyze the large yellow croaker matrix from the tartrazine extracts in the large yellow croaker peel experiment. Figure 5a,b show more impure peaks, such as the peak at 1428 cm^−1^, than in Figure 4 due to the interference of non-targeted components. For the first preparation of the W sample, an approximate 12% decrease was observed at the intensity peak of 1599 cm^−1^ in Figure 5a compared with that in Figure 5b, thereby indicating that the recovery rate of the large yellow croaker was approximately 88% of the sample preparation. Similarly, the recovery rates of the preparation of the W+M and W+M+C samples were 84% and 81%, respectively. From W to W+M+C, the decrease in recovery rate may have been due to the increase in purification steps. As shown in Figure 5b, W+M+C had a significant increase in SERS signal, although it had more purification steps than the afore mentioned methods. The W+M+C method took less than 20 mins to prepare the sample, the decrease in recovery rate was minimal and the method showed positive SERS performance. Therefore, this sample preparation method was chosen as the optimal method for further tartrazine detection in large yellow croaker.

Tartrazine is used illegally to color fish to make low-priced fish, such as albiflora croaker, appear similar to high-priced yellow croaker or putrescent yellow croaker. Dripping standard tartrazine solutions on a white paper showed that 2.5 ppm was the lowest critical concentration identified by the naked eye, converting 25.48 ng/cm^2^ to 200 μL in an area of 19.62 cm^2^. Hence, the color was completely invisibility when this concentration was added to yellow croaker peels.

We detected tartrazine on the basis of the optimal sample preparation. Figure 6 illustrates the SERS spectra obtained from seven different tartrazine concentrations adsorbed in the large yellow croaker. The spectra revealed that the lowest detectable concentration was 20.38 ng/cm^2^, which was lower than 25.48 ng/cm^2^. Thus, the as-prepared Ag nanowires coupled with SERS technology can meet the requirement of zero-tolerance residue limit. Compared to the HPLC method, this method has the advantage of a short detection time. However, there is still a long way to go before the SERS technique overcomes the matrix interference absolutely.

## 4. Conclusions

In this study, the as-prepared Ag nanowire SERS substrate was tested as a potential tool to determine tartrazine in large yellow croaker for the first time. Highly uniform and high-quality Ag nanowires were synthesized in a time-efficient and cost-effective manner, which displayed the substantially enhanced effect on standard tartrazine solution detection. The results showed that the lowest detectable concentration was 10 ng/mL. The PLS model demonstrated good predictability. In terms of recovery rate, time, cost and SERS signal, the optimal sample preparation method (i.e., W+M+C) was selected and could be detected at concentration levels as low as 25.48 ng/cm^2^, which fully satisfies the actual detection requirements. Hence, the established SERS approach showed remarkable performance in determining tartrazine residue in large yellow croaker.

## Figures and Tables

**Figure 1 nanomaterials-08-00967-f001:**
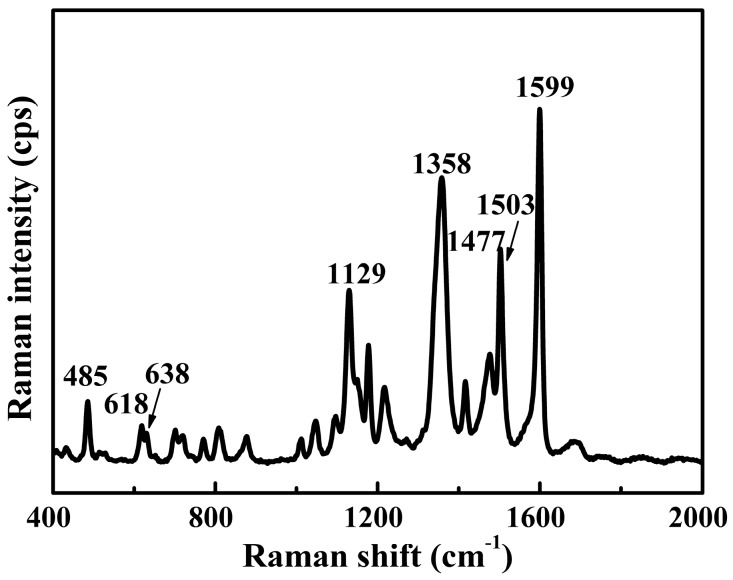
Raman spectrum of tartrazine.

**Figure 2 nanomaterials-08-00967-f002:**
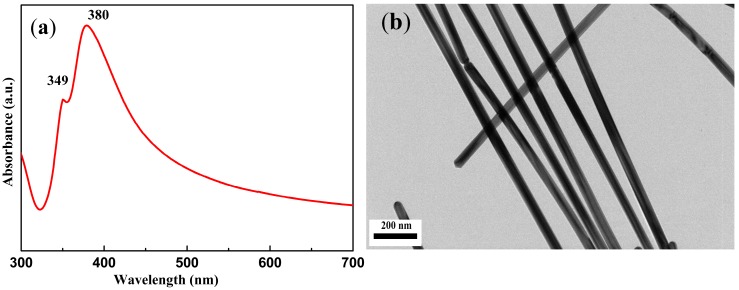
(**a**) Ultraviolet-visible (UV-Vis) absorbance spectrum of Ag nanowires; (**b**) transmission electron microscopy (TEM) images of Ag nanowires.

**Figure 3 nanomaterials-08-00967-f003:**
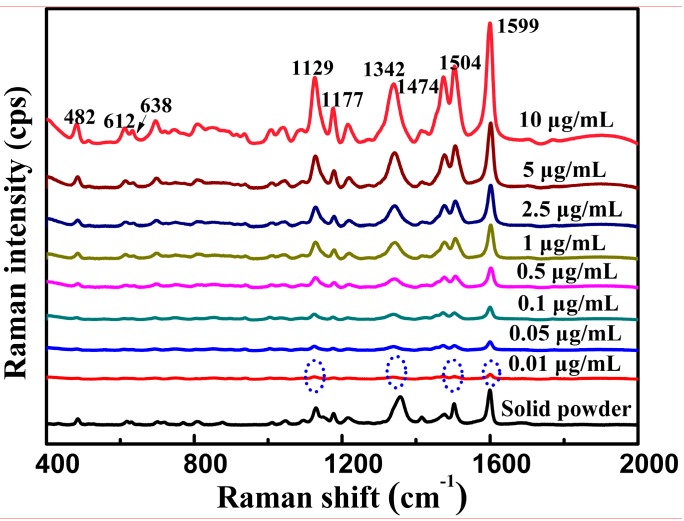
Surface-enhanced Raman scattering (SERS) optical spectra of the standard tartrazine solution (*n* = 40).

**Figure 4 nanomaterials-08-00967-f004:**
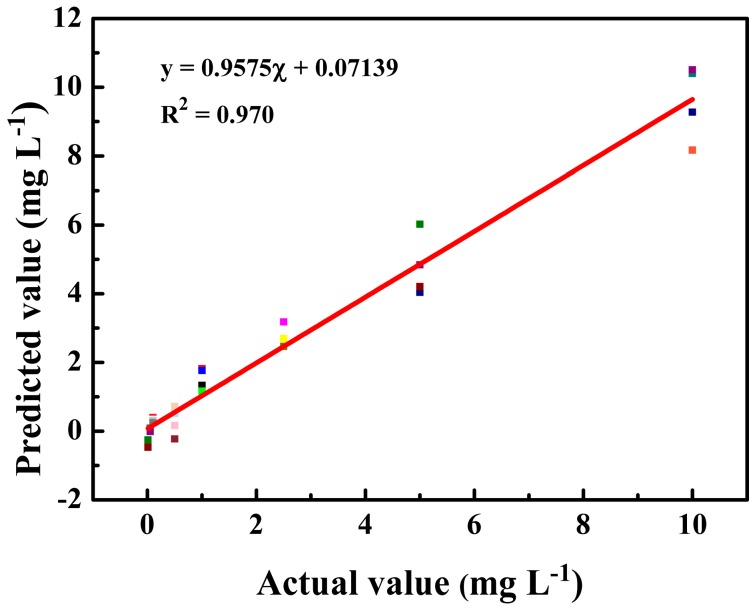
Partial least square (PLS) model of the standard tartrazine solution.

**Figure 5 nanomaterials-08-00967-f005:**
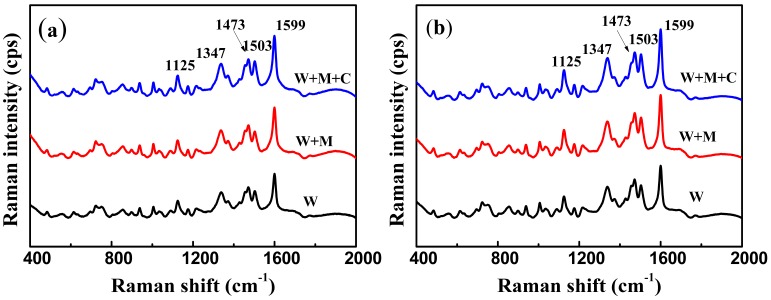
(**a**) Surface-enhanced Raman scattering (SERS) spectra extract from large yellow croaker peels spiked with 509.55 ng/cm^2^ of tartrazine; (**b**) extract added with tartrazine equivalent (a) but assumed 100% recovery rate; (**c**) Raman intensity of tartrazine at 1599 cm^−1^ via different sample preparation methods (*n* = 20).

**Figure 6 nanomaterials-08-00967-f006:**
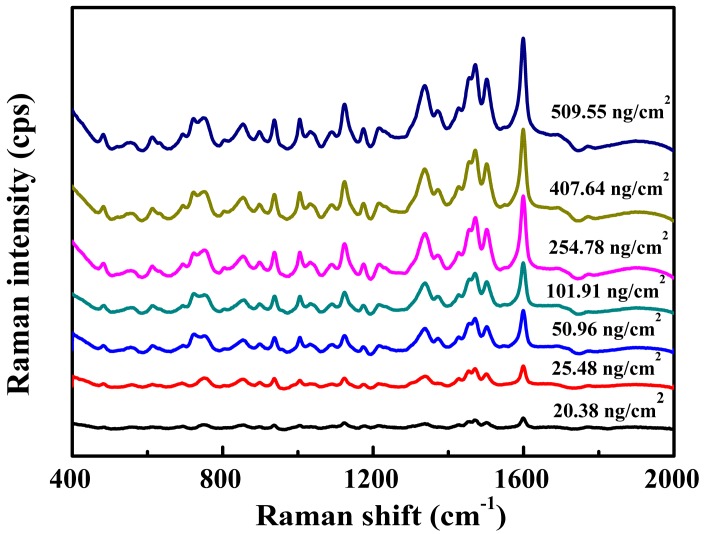
Representative surface-enhanced Raman scattering (SERS) spectra of tartrazine extracts from large yellow croaker skin (*n* = 30).

**Table 1 nanomaterials-08-00967-t001:** Major band assignment of Raman spectra for tartrazine.

Raman Shift (cm^−1^)	Vibrational Assignment
485	ρ (SO_3_^−^), Def_op_ (C–H)_ph1, pyr_
618	Def_op_ (C–H)_ph1, ph2_, δ (pyr)
638	Def_op_ (ph1, ph2)
1129	Def_op_ (C–H)_ph1, ph2,_ δ (ph1)
1358	υ (–C–N=N–C–), υ_s_ (COO^−^), ρ (ph2)
1477	δ (C–H)_ph1, ph2,_ υ_as_ (N=N–C)
1503	δ (ph2), δ (C=C)
1599	υ (ph1, ph2), Def_op_ (C–H)_ph1, ph2_, δ (OH), υ_as_ (COO^−^)

υ_s_—symmetric stretching; υ_as_—asymmetric stretching; δ—in-plane bending; γ—out-of-plane bending; ip—in-plane; op—out-of-plane; ρ—rocking; ph—phenyl ring; pyr—pyrazole ring; Def—deformation.

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
