# Peer review of "Rapid Tartrazine Determination in Large Yellow Croaker with Ag Nanowires Using Surface-Enhanced Raman Spectroscopy"

_nanomaterials, 2018, doi:10.3390/nano8120967_

Round 1
Reviewer 1 Report
After reading the paper entitled "Rapid tartrazine determination in large yellow croaker by surface-enhanced Raman spectroscopy coupled with Ag nanowires", this article sounds interesting. However, in its present form, this paper is not acceptable for publication. Major revisions should be addressed before reconsidering my decision.
1-Please, can you change your title by "Rapid tartrazine determination in large yellow croaker with Ag nanowires by using surface-enhanced Raman spectroscopy"? Because your actual title is not clear, it is the word "coupled" which is not suitable in your case, I think.
2-I strongly recommend to add the absorption or extinction of your Ag nanowires and also to calculate the enhancement factor in order to explain the SERS effect observed in your experiments. Indeed, it misses this part of discussion. Thus, this additional part strengthens your paper.
3-Please read 2 or 3 times your paper, because it remains some spelling errors like in page 3 line 126 "by calculating 100 nanoparticles", I think that is "nanowires" and not "nanoparticles"
Author Response
Review 1
After reading the paper entitled "Rapid tartrazine determination in large yellow croaker by surface-enhanced Raman spectroscopy coupled with Ag nanowires", this article sounds interesting. However, in its present form, this paper is not acceptable for publication. Major revisions should be addressed before reconsidering my decision.
1. Please, can you change your title by "Rapid tartrazine determination in large yellow croaker with Ag nanowires by using surface-enhanced Raman spectroscopy"? Because your actual title is not clear, it is the word "coupled" which is not suitable in your case, I think.
Revised as per the suggestion, the title of this article has been modified as "Rapid tartrazine determination in large yellow croaker with Ag nanowires by using surface-enhanced Raman spectroscopy".
2. I strongly recommend to add the absorption or extinction of your Ag nanowires and also to calculate the enhancement factor in order to explain the SERS effect observed in your experiments. Indeed, it misses this part of discussion. Thus, this additional part strengthens your paper.
Revised as per the suggestion, related content was added in the revised manuscript in page 2, Line 78-80; page 4, Line 135-140; page 5, Line 156-159.
3. Please read 2 or 3 times your paper, because it remains some spelling errors like in page 3 line 126 "by calculating 100 nanoparticles", I think that is "nanowires" and not "nanoparticles"
Revised in page 4, Line 140. Moreover, this paper has been carefully inspected, spelling and other problems have been corrected and marked.
Reviewer 2 Report
The manuscript describes utilization of surface-enhanced Raman spectroscopy (SERS) technology coupled with Ag nanowires to detect tartrazine in large yellow croaker. The manuscript is interesting. However, the use of SERS for detecting tartrazine has already been reported in the following literatures. For example, in, Ai et al, Rapid qualitative and quantitative determination of food colorants by both Raman spectra and Surface-enhanced Raman Scattering (SERS), Food Chemistry, Volume 241, 15 February 2018, Pages 427-433; combined Surface Enhanced Raman Spectroscopy (SERS)/UVâ vis approach for the investigation of dye content in commercial felt tip pens inks, S Maddalena, T Abeer, A Antonio, M Rodorico, G Piero - Talanta, 2018; Handheld surface‐enhanced Raman scattering identification of dye chemical composition in felt‐tip pen drawings, Daniela Saviello Alexandra Di Gioia Pierre‐Ives Turenne Maddalena Trabace Rodorico Giorgi Antonio Mirabile Piero Baglioni Daniela Iacopino, 2018, https://doi.org/10.1002/jrs.5411. Therefore, the novelty is damped, although application in yellow croaker is a new development.
Some specific comments are:
1. What is the average enhancement factor of the nanowire? More details are required, especially, the authors need to explain how the number of molecules such as NREF and NSERS were calculated? It is very important to obtain a reliable value of number of molecules contributing to the normal Raman signal and correctly measure the reliable Raman intensity I(NRS) contributed by that number of molecules.
2. What is the actual improvement in signal intensity in SERS compared to normal Raman?
3. Does the yellow croaker extracted sample has any fluorescence background? How do you eliminate all the interference from protein and DNA?
4. Since Ag nanowire will be random shape and size, what is the uniformity (variance) of the SERS signal? How does it vary sample to sample.
5. What is the limit of detection (LOD) of this method? How does it compare to HPLC?
6. The reliability and uncertainty of this method should be compared to gold standard 9such as HPLC) methods.
7. The concentrations are presented in ng/cm2, how will it be in ng/kg?
Author Response
Review 2
The manuscript describes utilization of surface-enhanced Raman spectroscopy (SERS) technology coupled with Ag nanowires to detect tartrazine in large yellow croaker. The manuscript is interesting. However, the use of SERS for detecting tartrazine has already been reported in the following literatures. For example, in, Ai et al, Rapid qualitative and quantitative determination of food colorants by both Raman spectra and Surface-enhanced Raman Scattering (SERS), Food Chemistry, Volume 241, 15 February 2018, Pages 427-433; combined Surface Enhanced Raman Spectroscopy (SERS)/UVâ vis approach for the investigation of dye content in commercial felt tip pens inks, S Maddalena, T Abeer, A Antonio, M Rodorico, G Piero - Talanta, 2018; Handheld surface‐enhanced Raman scattering identification of dye chemical composition in felt‐tip pen drawings, Daniela Saviello Alexandra Di Gioia Pierre‐Ives Turenne Maddalena Trabace Rodorico Giorgi Antonio Mirabile Piero Baglioni Daniela Iacopino, 2018, https://doi.org/10.1002/jrs.5411. Therefore, the novelty is damped, although application in yellow croaker is a new development.
References of using SERS for detecting tartrazine were added in the revised manuscript as per the suggestion in page 2, Line 52-53.
1. What is the average enhancement factor of the nanowire? More details are required, especially, the authors need to explain how the number of molecules such as NREF and NSERS were calculated? It is very important to obtain a reliable value of number of molecules contributing to the normal Raman signal and correctly measure the reliable Raman intensity I(NRS) contributed by that number of molecules.
The following equations defined enhancement factor (EF) and analytical enhancement factor (AEF).
EF is the most significant indicator to the sensitivity of SERS substrate. While in the actual application, it is difficult to calculate the number of molecules in the normal Raman measurement (NREF) and the number of absorbed molecules in the SERS measurement (NSERS). To obtain more simple and easily-measured method, AEF is preferable, CNRS and CSERS are the molecular concentration in the normal Raman and SERS measurement respectively.
The enhancement factor for tartrazine was 3.9×106. Some related content was added in revised manuscript in page 5, Line 156-159.
2. What is the actual improvement in signal intensity in SERS compared to normal Raman?
In our study, Raman spectrum of tartrazine solid was collected, the intensity of the characteristic peak around 1599 cm−1 of Raman spectrum of tartrazine solid was close to the SERS spectra of 1 μg/mL tartrazine standard solution (Figure 3), indicating the Raman scattering signals improved greatly in SERS technique. Content about enhancement factor was added in revised manuscript in page 5, Line 156-159.
3. Does the yellow croaker extracted sample has any fluorescence background? How do you eliminate all the interference from protein and DNA?
In terms of the selection of sample, we chose the yellow croaker skin instead of yellow croaker meat to avoid the more complex effect from matrix. But the yellow croaker skin extract did has fluorescence background even under the optimal sample preparation, which can clearly found from the decrease intensity of the characteristic peak around 1599 cm−1 and unstable baseline.
4. Since Ag nanowire will be random shape and size, what is the uniformity (variance) of the SERS signal? How does it vary sample to sample.
To address the reviewer’s concern about the reproducibility, in our previous studies[29] we calculated the relative standard deviation (RSD) values of Raman intensity based on three characteristic brands of the SERS spectra of 0.1 ng/mL CV collected with Ag NWs substrates from different batches and the RSD values of Raman intensity at three different peaks were all below 15%.
5. What is the limit of detection (LOD) of this method? How does it compare to HPLC?
The limit of detection of this method is 0.01 μg/mL, which is about two times of HPLC. Some related content was added in page 5, Line 146-149.
6. The reliability and uncertainty of this method should be compared to gold standard (such as HPLC) methods."
The related information was added on page 5, Line 146-149, and page 7, Line 202-206 as per the suggestion
7. The concentrations are presented in ng/cm2, how will it be in ng/kg?
Large yellow croakers are often colored by soaking in tartrazine solution, so the coloring sites are the yellow croakers’ peels not the meat, it is closer to actual samples to measure the area of large yellow coakers, thus we don’t change in ng/kg during our research.

Round 2
Reviewer 1 Report
The authors have made the corrections asked by the reviewers and it can be accepted for publication in Nanomaterials.
Reviewer 2 Report
The authors have revised the manuscript according to comments.